# Effects of Nutritional Plane at Breeding on Feed Intake, Body Weight, Condition Score, Mass Indexes, and Chemical Composition, and Reproductive Performance of Hair Sheep

**DOI:** 10.3390/ani13040735

**Published:** 2023-02-18

**Authors:** Raquel Vasconcelos Lourencon, Amlan Kumar Patra, Ryszard Puchala, Lionel James Dawson, Luana Paula dos Santos Ribeiro, Fabiola Encinas, Arthur Louis Goetsch

**Affiliations:** 1American Institute for Goat Research, Langston University, Langston, OK 73050, USA; 2College of Veterinary Medicine, Oklahoma State University, Stillwater, OK 74078, USA

**Keywords:** body composition, body mass index, hair sheep, nutritional plane, reproductive performance, supplementation

## Abstract

**Simple Summary:**

A high nutritional plane before and during breeding may improve reproductive performance of hair sheep fed low quality forage. Supplementation of Dorper, Katahdin, and St. Croix sheep consuming wheat straw ad libitum with soybean meal at 0.16% body weight (BW) or a mixture of 25% soybean meal and 75% ground corn at 0.8% BW (dry matter basis) for 8 weeks before and in the early breeding period caused differences in BW, body condition score, and body composition. Supplementation with soybean meal alone increased wheat straw intake, resulting in similar total feed intake relative to BW. Reproductive performance was not affected by supplement treatment, which may be attributable to the moderate body condition score (3 to 3.5 on a 5-point scale) before breeding and the difference in wheat straw intake. Also, some results indicate that body mass indexes can be better predictors of change of body composition compared with body condition score.

**Abstract:**

This study was conducted to assess effects of the nutritional plane before and during breeding on feed intake, body weight (BW) change, reproductive performance, body condition score (BCS), body mass indexes (BMI), and body composition of three breeds of hair sheep. Twenty-five Dorper, 27 Katahdin, and 33 St. Croix ewes were allocated to groups and treatments based on initial BW and age. Supplementation treatments were soybean meal fed at 0.16% BW (LS) and a mixture of 25% soybean meal and 75% ground corn given at 0.8% BW (HS; dry matter basis) for 88 to 102 days including 17 days after breeding. Wheat straw intake (% BW) was greater (*p* < 0.01) for LS than for HS. Average daily gain and BCS change were similar among breeds, but both were greater for HS vs. LS. Birth rate, litter size, total litter birth weight, gestation length, and number of services were similar among breeds (*p* > 0.05), although individual lamb birth weight was lower for St. Croix than for Katahdin (*p* < 0.05). None of these reproductive variables were influenced by treatment or the breed × treatment interaction (*p* > 0.05). There were no breed differences in whole body concentrations of water, fat, protein, ash, or energy in week 0 or 4, but in 8 week concentrations of water, protein, and ash were greatest among breeds for St. Croix (*p* < 0.05) and levels of fat and energy were lower for St. Croix than for Dorper (*p* < 0.05). In week 8, total amounts and concentrations of fat and energy were greater for HS vs. LS and those of water, protein, and ash were less for HS (*p* < 0.05). There were interactions (*p* < 0.05) between treatment and period in blood concentrations of cholesterol and non-esterified fatty acids and between breed, treatment, and period in level of triglycerides and total antioxidant capacity. The concentration of cortisol was greater for HS vs. LS (*p* < 0.05). In conclusion, supplementation did not influence reproductive performance despite differences in BW and BCS change, which may be due to the initial moderate BCS and greater wheat straw intake for the LS treatment.

## 1. Introduction

Ruminant production systems typically are based on relatively low-cost and -quality forages or roughages, including crop byproducts that are typically deficient in crude protein, minerals, and available energy, particularly during dry seasons. Such feedstuffs alone cannot optimize ruminal microbial growth and, as a consequence, fulfill nutritional demands of ruminants with moderate to high requirements [1]. Therefore, in many instances feedstuffs like crop residues are supplemented with concentrate feedstuffs to achieve acceptable levels of feed intake and animal performance [1,2].

It can be useful to periodically determine body weight (BW) and evaluate indicators of body composition to assess the appropriateness of the level and composition of a supplement, with possible modifications to achieve target conditions of growth, reproductive performance, and milk production. An increased plane of nutrition before and during the breeding season sometimes improves reproductive performance in sheep [3,4]. For example, diet modifications to increase amino acid availability can improve oocyte quality during antral follicular development and favorably impact fetal and placental metabolism [5]. Also, energy intake and body energy stores can regulate secretion of gonadotropin releasing hormone/luteinizing hormones by modulating related hypothalamic gene expressions and subsequently follicular development and ovulation rate [6,7].

One common means of predicting body composition is body condition score (BCS), as addressed for goats by Ngwa et al. [8] and Sanson et al. [9] and Morel et al. [10] with sheep. However, because BCS is subjective, there has been interest in use of more objective measures, including body mass index (BMI). Most often a BMI is a ratio of BW to one or more linear measure or expressions (e.g., Tanaka et al. [11]; Dønnem et al. [12]; Randby et al. [13]; Vilar-Martínez et al. [14]; Liu et al. [15]). There have not been particular BMI identified as preferable relative to others in terms of relationships with present or changes in body composition. Usually the linear measures are planar, but results of Eknæs et al. [16] with multiparous Norwegian dairy goats during a 230-day lactation period provide some support for consideration of a volumetric measure in that BW and BCS changed whereas BMI based on planar measures did not. Other than study of relationships between BMI and specific indices of milk and meat productivity of Wallachian sheep by Ptáček et al. [17], there does not appear to have attention given to relationships of various BMI to changes in body composition. A factor limiting progress in this area is relating measures to actual whole-body composition. A method such as urea dilution or urea space offers a relatively simple means not requiring slaughter, which would be of particular interest regarding changes over time rather than static values.

Hair sheep have increased in popularity in the USA, which relates to the low value of wool and their hardiness compared with wool breeds [18]. The three most common hair sheep breeds in the USA are Dorper, Katahdin, and St. Croix [19], which differ in body structure and conformation, BW, reproductive performance, susceptibility to internal parasitism, and resilience to stressors such as high heat load conditions and limited drinking water availability [18,20,21,22,23]. But, responses to varying nutritional planes at critical times such as the breeding season have not been extensively studied, as is also the case for how measures predictive of body composition relate to reproductive performance of hair sheep breeds. Therefore, objectives of this experiment were to evaluate effects of the nutritional plane before and during breeding on feed intake, BCS, various indicators of body composition, and reproductive performance of Dorper, Katahdin, and St. Croix ewes.

## 2. Materials and Methods

### 2.1. Animals

The protocol for the experiment was approved by the Langston University Animal Care and Use Committee. The study occurred in the late Summer of 2018 (10 September) and through lambing in the Spring of 2019. There were 85 female hair sheep used, 25 Dorper, 27 Katahdin, and 33 St. Croix with mostly multiparous (only two Katahdin, one Dorper and one St. Croix sheep were primiparous). Before the experiment, sheep were vaccinated against clostridial organisms, and ones with FAMACHA© score greater than 3 were treated for internal parasites. Initial age when the study began and the different supplement treatments noted below were imposed averaged 4.9 yr (SEM = 0.20) and ranged from 2.8 to 11.1 yr.

### 2.2. Treatments, Housing, and Breeding

The sheep were allocated to four groups per breed based on initial BW and age, with six to nine animals per group, and two supplement treatments were assigned to the groups within breed (i.e., two groups per breed and supplement; Figure 1). There were 12 pens (four for each breed) and each group was housed in a pen with two sections, a 3.66 m × 3.66 m area within a building and a second with the same dimensions located outside but also covered by a roof. The floors were concrete and periodically cleaned. Water was available ad libitum in automatic waterers and there was free access to salt blocks. Artificial lighting was provided from 07:00 to 16:00. There were two feed troughs per pen in the front area, one 2.44 m in length and another 1.22 m, which was sufficient to avoid competition among and aggressive behaviors of animals for feed access. There were six pens on one side of the facility (set 1) and six on the other side (set 2), with a hallway between the feed troughs and pens. There was one breed × supplement treatment group on each side of the facility.

All animals began the experiment at the same time, but because there was a separate breeding period for the two animal sets, lengths of time supplement treatments were imposed differed slightly (Figure 2). For many of the measures, data were collected at the beginning of the experiment and end of periods 1 and 2 that were 4 week (wk) in length. Supplement treatments in periods 1 and 2 as well as period 3 were soybean meal fed at approximately 0.16% of initial BW (LS) and a mixture of 25% soybean meal and 75% ground corn given at approximately 0.8% of initial BW (HS; dry matter basis). During the 133 days sheep were in the pens before being moved outside on pasture, coarsely ground wheat straw was offered at approximately 120% of consumption on the preceding few days and was consumed ad libitum. After straw refusals were collected and weighed, supplements were fed at 08:00, which were completely consumed, and then wheat straw was given. Wheat straw and supplement samples were collected weekly, ground to pass a 1-mm screen, and analyzed for dry matter (DM), ash [24], crude protein (CP; Leco TruMac CN, St. Joseph, MI, USA), and neutral detergent fiber (NDF) with use of heat stable amylase and containing residual ash ([25]; filter bag technique of ANKOM Technology Corp., Fairport, NY, USA).

Period 3 was 32 and 46 days for sets 1 and 2, respectively. Hence, supplement treatments were imposed for 88 and 102 days for sets 1 and 2, respectively. Breeding began on days 71 and 85 for sets 1 and 2 and, thus, supplement treatments continued for 17 days after the breeding period was initiated. In period 4, all groups were given the HS supplement treatment regime, which was 45 and 31 days for sets 1 and 2, respectively. The total breeding period was 34 days, with two 17-day cycles. As noted above, the two supplement treatments were imposed for the first cycle, whereas all animals were switched to the HS supplement treatment during the second cycle. Four rams of each breed divided into two sets were used, which were previously subjected to and passed a breeding soundness examination. Estrus synchronization was employed, with a controlled internal drug release (CIDR) vaginal device (EAZI-breed^™^ CIDR^®^, Pfizer Animal Health, Auckland, New Zealand) inserted on day 0 (on day 61 for set 1 and day 75 for set 2 after supplementation started) of estrus synchronization and 10 mg of PGF2α (Lutalyse, 10 mg dinoprost tromethamine i.m., Zoetis Animal Health, Parsippany-Troy Hills, NJ, USA) was intramuscularly injected 9 days later (on day 70 for set 1 and day 84 for set 2 after supplementation started). On day 10 (on day 71 for set 1 and day 85 for set 2 after supplementation started), the CIDR was removed, at which time two rams of each breed fitted with marking harnesses were introduced into one of the two pens per breed-supplement treatment. Estrus and breeding markings were checked twice daily at 08:00 and 16:00. Non-return to estrus was assumed when ewes were not marked during the second cycle with a ram. After 17 days with the first group of ewes, the rams were moved with the second group and stayed with them for two cycles. The second set of two rams was used for the second cycle of the first group of ewes. Ultrasound pregnancy diagnosis was performed at 25- and 40-days post-breeding.

The number of embryos and presence/absence of luteal structures was determined using transrectal ultrasound imaging on day 25 post-breeding as described previously by Schrick and Inskeep [26]. Transrectal ultrasonography was performed with a 7.5 MHz, linear-array transrectal transducer connected to a portable Aloka 500 SSD-500V (Aloka Co. Ltd., Tokyo, Japan) in accordance with techniques described by Meinecke-Tillmann [27]. All animals that were pregnant on day 25 were confirmed pregnant on day 40 by ventral external examination with a 3.5 MHz linear-array transducer. The intent was to also evaluate the number of embryos on day 40 to address embryo mortality, but this was not achieved. Other reproduction data of number of services per conception, birth rate, gestation length, litter size, individual lamb birth weight, and litter birth weight were determined at lambing time, which occurred in portable lambing pens. Gestation length was based on the last day of observed breeding and the date of birth.

After period 4 the sheep were moved outside on pastures and managed similarly, consuming grass hay ad libitum and receiving a concentrate-based supplement in an increasing amount as the gestation period advanced. Approximately 1 month before lambing, ewes were vaccinated again against clostridial organisms, and ones with FAMACHA© score greater than 3 were treated for internal parasites. Ewes were moved back to the same facility approximately 1 wk before lambing.

### 2.3. Body Condition Score and Body Mass Indexes

In addition to full BW, unshrunk BW was determined before feeding at the beginning and end of periods 1, 2, and 3. At the beginning of period 1 and end of periods 1 and 2, a number of variables were evaluated. Body condition score (BCS) was assessed as described by Ngwa et al. [8] in a 5-point scale and there were linear measures of height at the withers (Wither), length from the point of the shoulder to the hook bone (Hook) and pin bone (Pin), and circumference from heart girth (Heart). Of the 13 BMI described by Liu et al. [15], four most meaningful as recommended by Wang et al. [28] were estimated as noted below.
BMI-WH = BW/(Wither × Hook) [g/cm^2^]
BMI-WP = BW/(Wither × Pin) [g/cm^2^]
BMI-GH = BW/(Heart × Hook) [g/cm^2^]
BMI-GP = BW/(Heart × Pin) [g/cm^2^]

### 2.4. Blood Constituents

Blood was sampled at the end of periods 1 and 2 in the morning when measures noted above were made, and samples were also taken after period 3. Samples were collected by jugular venipuncture into two tubes used for plasma, one with sodium fluoride and potassium oxalate and another with sodium heparin, and a third tube without an anticoagulant for serum. Plasma and serum were harvested by centrifugation for 20 min at 3000× *g* and frozen at −20 °C. Plasma from the sodium fluoride and potassium oxalate tube was analyzed for glucose and lactate, respectively, with a USI 2300 Plus Glucose & Lactate Analyzer (YSI Inc., Yellow Springs, OH, USA). Plasma from the sodium heparin tubes and (or) serum was analyzed for constituents such as non-esterified fatty acids (NEFA), triglycerides, cholesterol, urea, albumin, and total protein with a Vet Axcel^®^ Chemistry Analyzer (Alfa Wassermann Diagnostic Technologies, West Caldwell, NJ, USA) according to the manufacturer’s instructions. Plasma was analyzed for total antioxidant capacity (TAC) colorimetrically with a Technicon Autoanalyzer II System (Technicon Instruments, Tarrytown, NY, USA) based on a ferric reducing ability of plasma [29].

### 2.5. Body Composition and Heart Rate

The urea space or dilution method was used to estimate body composition as done previously in sheep [30,31,32,33]. After blood samples addressed above were collected, feed and water were withheld for 24 h and shrunk or empty BW (EBW) was determined. A winged catheter (19 gauge; Vacutainer^®^, Becton Dickinson, Rutherford, NJ, USA) was placed in a jugular vein and used to collect a 0-min sample into a tube without a coagulant and then to infuse a sterilized 20% (wt/vol) urea solution in 0.9% (wt/vol) saline at 130 mg/kg BW. A blood sample was collected 12 min after the mid-point of the infusion. Serum was harvested by centrifuging at 1500× *g* and 4 °C for 15 min. However, there were some missing observations at the beginning of periods 1 and 2 due to problems with infusion or sampling procedures. Urea space (USP) was calculated as urea infused/change in serum concentration before infusion and at 12 min. Water and fat in the EBW (EBH_2_O and EBFAT, respectively) were calculated with regressions presented by Poland [34]. The equation for kg EBH_2_O was 7.86 + (0.259 × EBW in kg) + (0.195 × USP in kg), and that for kg EBFAT was −8.92 + (0.625 × EBW in kg) − (0.275 × USP in kg). Percentages of protein and ash in the empty body (% EBPRO and % EBASH, respectively) were estimated as described by Reid et al. [35]. The equation for % EBPRO was % EBH_2_O × 0.27173, and that for % EBASH is % EBH_2_O × 0.06356. To determine energy accretion, 5.52 and 9.4 kcal/g (23.096 and 39.330 kJ/g) of protein and fat, respectively, were assumed [36].

Heart rate (HR) was measured in the third week of periods 1 and 2 as described by Puchala et al. [37,38]. Heart rate was determined for 24 h with animals in four pens at a time or a total period of 3 days. Sheep were fitted with stick-on ECG electrodes (Cleartrace, Utica, NY, USA) attached to the chest just behind and slightly below the left elbow and at the base of the jugular groove on the right side of the neck. Electrodes were secured to skin with a 5 cm wide elastic bandage (Henry Schein, Melville, NY, USA) and animal tag cement (Ruscoe, Akron, OH, USA). There were ECG snap connecting leads (Bioconnect, San Diego, CA, USA) used to connect electrodes to T61coded transmitters (Polar, Lake Success, NY, USA). Human S610 heart rate (Polar) monitors with wireless connection to the transmitters were used to collect HR data at a 1-min interval for 24 h. Heart rate data were analyzed using Polar Precision Performance SW software.

### 2.6. Statistical Analyses

The normality of data distribution for most important variables was evaluated and verified with the Shapiro-Wilk statistic of the Univariate procedure of SAS [39]. Most data were analyzed with the MIXED procedure of SAS [39]. Different covariance structures were compared via Akaike’s Information Criterion, but values were lower for Variance Components or differences were not marked. Animal group or pen within the breed and supplement treatment was the experimental unit. For the reproduction variables of birth rate, litter size, and number of services, the analysis was performed with the GLIMMIX procedure of SAS. Interaction means are presented regardless of significance of the interaction. Differences among means were determined by least significant difference with a protected F-test (*p* < 0.05).

## 3. Results

### 3.1. Feed Composition and Intake

The high levels of NDF and ADL in wheat straw reflect the low quality of the roughage source and limited digestibility (Table 1). However, the level of CP was slightly greater than expected based on data compiled by Preston [40]. As expected, the composition of supplements differed considerably in levels of constituents such as CP.

There were no interactions (*p* > 0.05) in feed intake measures between breed and supplementation treatment as well as for others except three-way interactions noted later for two blood constituent concentrations determined in different periods. The only difference among breeds in wheat straw and total DM intake in % BW was in period 1, with greater values for STC than for DOR and KAT (*p* = 0.031; Table 2). However, there also were similar trends in period 2 (*p* < 0.07). Wheat straw DM intake in % BW was greater (*p* < 0.05) for the LS than HS supplementation treatment in each period. As a result, total DM intake in % BW was not different between supplement treatments in periods 1, 2, and 3 (*p* > 0.05). Wheat straw DM intake in g/day was greater for LS vs. HS in period 4 (*p* < 0.020), although the difference was less than earlier (i.e., differences of 28.7, 18.5, 23.3, and 11.3% in periods 1, 2, 3, and 4, respectively). Though BW data are not available, total DM intake in period 4 relative to BW at the end of period 3 was actually greater (*p* = 0.026) for LS vs. HS (2.58, 2.20, 2.68, 2.40, 2.73, and 2.48% BW for DOR-LS, DOR-HS, KAT-LS, KAT-HS, STC-LS, and STC-HS, respectively; SEM = 0.127).

### 3.2. Body Weight and Composition

Naturally there were differences among breeds in BW and mass of water, fat, protein, ash, and energy in wk 0, 4, and 8 (*p* < 0.05; Table 3). There were no breed differences in concentrations of water, fat, protein, ash, or energy in wk 0 or 4, but in 8 wk concentrations of water, protein, and ash were greatest among breeds for STC (*p* < 0.05) and levels of fat and energy were lower for STC than for DOR (*p* < 0.05). Supplementation treatment did not affect (*p* > 0.05) total quantities and concentrations of water, fat, protein, ash, or energy in wk 4, but in wk 8 total amounts and concentrations of fat and energy were greater for HS vs. LS and those of water, protein, and ash were less for HS (*p* < 0.05).

There were no breed effects on change in BW or quantities of water, fat, protein, ash, or energy in wk 0 to 4, 4 to 8, or 0 to 8 (*p* > 0.05; Table 3). Changes in BW and quantities of water, fat, protein, ash, and energy from 0 to 4 wk were greater for HS vs. LS, but from wk 4 to 8 only change in full BW differed between supplementation treatments (*p* < 0.05). As a result of differences from wk 0 to 4, change in BW and quantities of water, fat, protein, ash, and energy from 0 to 8 wk were also greater for High vs. Low (*p* < 0.05).

### 3.3. Reproductive Performance

Birth rate, litter size, total litter birth weight, gestation length, and number of services were similar among breeds (*p* > 0.05), although individual lamb birth weight was lower for STC than for KAT (*p* < 0.05; Table 4). None of these variables were influenced by supplementation treatment or the breed × supplementation treatment interaction (*p* > 0.05).

### 3.4. Heart Rate and Blood Constituents

Heart rate and blood glucose concentration were not influenced by breed, supplementation treatment, period, or any interaction thereof (*p* > 0.05; Table 5). There were many effects of period (*p* < 0.05), including concentrations of lactate, total protein, albumin, cholesterol, triglycerides, NEFA, cortisol and TAC in blood. Moreover, there were supplementation treatment × period interactions for cholesterol, TAC, and NEFA and a three-way interaction involving breed, supplementation treatment, and period for triglyceride concentration and TAC (*p* < 0.05).

The lactate concentration was greater in period 1 than in 2 and 3 (*p* < 0.05; Table 6). The concentration of total protein ranked (*p* < 0.05) period 1 > 2 > 3, and the concentration of albumin was lowest among periods in period 1 (*p* < 0.05). With the LS supplementation treatment, the cholesterol level ranked (*p* < 0.05) period 1 > 2 > 3, whereas with the HS supplementation treatment the concentration was less in periods 2 and 3 vs. 1 (*p* < 0.05). Moreover, the cholesterol concentration was lower for HS vs. LS in periods 1 and 2 but similar between supplementation treatments in period 3 (*p* > 0.05). Overall, the triglyceride concentration ranked (*p* < 0.05) period 1 < 3 < 2, although magnitudes of change with advancing period varied among breed × supplementation treatments and values were similar among periods for KAT on LS. The NEFA concentration was lowest among periods in period 2 for LS and for HS was higher in period 1 than 2 and 3 (*p* < 0.05). Overall, the TAC was lowest among breeds for DOR (*p* < 0.05) and ranked (*p* < 0.05) period 1 < 2 < 3, although there was considerable variation among breed × supplementation treatment in change with advancing period. The level of cortisol was greater for the HS vs. LS supplementation treatment (*p* < 0.05) and in period 1 than 2 and 3 (*p* < 0.05).

### 3.5. BCS and BMI and Relationships with Body Composition

There were no differences among breeds in BCS in wk 0 and 4, but BCS in wk 8 was lower for STC than for DOR and KAT (*p* < 0.05; Table 7). All BMI in wk 0, 4, and 8 ranked (*p* < 0.05) STC < KAT < DOR except for BMI-GH that ranked STC and KAT < DOR. The only difference in change in these measures was for BCS from wk 4 to 8 (*p* < 0.05). The sole effect of supplementation treatment at the three times was on BCS in wk 8 and, thus, the change in BCS was greater from wk 0 to 4 and 8 for HS vs. LS (*p* < 0.05). There was just one significant change in BMI, which was for BMI-WH from wk 0 to 4.

All BMI were moderately to highly correlated (r = 0.59 to 0.71) with BCS at the three measurement times (*p* < 0.05; Table 8). Correlation coefficients between BMI and full and shrunk BW and concentrations of water, fat, protein, and energy were in most cases slightly higher than between BCS and these measures. However, there were many instances for which correlations between BCS and BMI-GH were similar. There were slightly stronger relationships for BMI-WP vs. BMI-WH and BMI-GP vs. BMI-GH at wk 0, 4 and 8.

Correlations between change in BCS and BMI-WH or BMI-WP were of moderate to low magnitude in different weeks (*p* < 0.051; Table 9). But correlations between change in BCS and BMI-GH or BMI-GP were not significant in all weeks. There were some correlation coefficients between BMI and full BW slightly greater than between BCS and full BW, although this did not include BMI-GH. Correlations between change in shrunk BW and BMI-WH and BMI-WP were greater than those for change in BCS from wk 0 to 4 and 0 to 8, although this was not the case for wk 4 to 8. There were no significant correlations between change in BMI from wk 0 to 4 or 4 to 8 and those in quantities of water, fat, protein, or energy, although all correlations for wk 0 to 8 were significant (*p* < 0.05). Conversely, there were some correlations for change in BMI from wk 0 to 4 and change in mass of water, fat, protein, and energy that were significant (*p* < 0.05), as was true for wk 4 to 8 but only for BMI-GH and BMI-GP regarding change in quantities of fat and energy. Similarly, all correlations between change in BMI and BW and quantities of water, fat, protein, and energy from wk 0 to 8 were significant, but only ones for BMI-WH and BMI-WP for change in fat and energy mass were greater than those between BCS and these measures.

## 4. Discussion

### 4.1. Feed Intake

Slightly greater wheat straw DM intake in % BW for STC relative to DOR and KAT is in accordance with findings of Tadesse et al. [23] and Hussein et al. [21]. Relatedly, Tadesse et al. [41] noted a corresponding small difference in the feed requirement for maintenance of STC than for DOR and KAT. Overall, the degree to which intake of wheat straw was greater for LS than for HS was somewhat unexpected. Intake by ruminants of low-quality forage diets is complex but primarily regulated by physical processing capacity, which encompasses ruminal digesta volume, passage rate, particle breakdown due to mastication during eating and rumination, and rate of microbial degradation. Although, metabolic conditions in regard to how nutrients and energy absorbed relate to and interact with requirements for the variety of physiological processes underway at any one given point in time have influence as well [42]. It has been generalized over the years that a dietary CP concentration of approximately 7% is necessary to avoid limited feed intake because of nitrogenous compound-related restrictions in one or more of the processes noted above [2,43,44]. That the CP concentration for LS was slightly greater than this presumably allowed for greater basal forage intake than realized with HS. Also, the relatively low amount of the LS supplement offered would have resulted in a greater difference between absorbed quantities of nutrients and energy and those required and (or) that could be potentially used in maintenance, tissue accretion, and reproductive processes. Another consideration is the likelihood of a negative feedstuff associative effect elicited by HS because of high levels of intake of corn and starch. A common generalization is that with supplemental concentrate less than 25–33% of total feed intake, negative associative effects do not occur or are minimal [45,46]. In the present study, the level of concentrate as a percentage of the total diet for HS was slightly greater than this threshold and, thus, could have decreased fiber digestion and (or) wheat straw intake. This is despite the increase in dietary CP concentration that otherwise would be stimulatory to intake and (or) digestion of a low-protein, low-quality roughage such as wheat straw. Assuming that BW at the end of period 4 did not markedly differ from that after period 3, greater total feed intake in period 4 for LS vs. HS probably was because of compensation in greater tissue accretion for the previous difference in nutritional plane, although this was achieved mainly by the higher level of supplement intake for LS animals in period 4 than earlier [47,48]. Heart rate was assessed because of its relationship with heat energy production although in this experiment it was not possible to determine the ratio of heat energy to HR in order to predict heat energy as in many other studies (e.g., Brassard et al. [49]; Keli et al. [50]; Silva et al., [51]). A tendency of increased HR for HS compared with LS might indicate a greater tendency of heat energy production in accordance with higher total feed intake in period 1 and 2.

### 4.2. Body Composition

The whole body concentration of protein was higher and that of fat lower for STC vs. DOR and KAT in wk 8, converse to similar values among breeds in wk 0 and 4. The latter finding could relate to the difference or tendency for a difference in wheat straw DM intake in periods 1 and 2. That is, typically with a similar plane of nutrition STC are leaner with a lower whole body fat concentration relative to DOR and KAT [18,19].

Higher whole-body mass of protein, fat, and energy for HS vs. LS at wk 8 relates to greater absorption presumably of both amino acids and energy for HS. However, the difference in composition at wk 8, with a higher level of fat and lower concentration of protein for HS, presumably relates primarily to differences in endproducts of digestion absorbed that affected hormonal conditions rather than being a function of total feed intake. That is, most likely with the HS supplementation treatment ruminal propionate production and absorption were greater than for LS, and perhaps there was some starch escaping ruminal digestion with intestinal absorption of glucose, which resulted in greater blood insulin levels to increase fat deposition [52,53].

### 4.3. Reproductive Performance

Elevated nutritional planes before and during breeding can enhance reproductive performance of sheep [4], probably as a result of an improved hormonal balance and follicular development [6,7]. In this study, restricted nutritional plane was subjected to 71 and 85 d before breeding started followed by high nutritional plane throughout the gestation period. Only one breed difference in reproductive measures suggests that each breed has the capacity and ability to perform well with moderate to low planes of nutrition. Also, as early embryonic death is an important factor for the litter size and birth weight [54,55], similar nutritional plane during most of the pregnancy period may result in similar reproduction performances in two groups. Similarly, no significant effects of supplementation treatment reflect an intent to have typical and practically relevant treatments rather than ones markedly different with very high and (or) low levels of reproductive performance. In this regard, very low and high BCS (e.g., 2.5 and 4.0, respectively) have adversely affected reproductive traits of sheep [56], with level of feed intake impacting conditions such as the absolute concentration and temporal fluctuations in luteinizing hormone [6]. In the present study, reproductive measures of sheep were not affected by supplementation treatment probably because of the optimal range of BCS of 3 to 3.5 before breeding, moderate impacts of supplementation treatments on feed intake and consequently nutritional status [57], and the elevated nutritional plane for all animals in the late gestation period. In a similar study, supplementation of 1% BW versus 0.15% BW continued up to third month of pregnancy improved reproduction performance in terms of litter size and litter birth weight [58].

### 4.4. Blood Constituents

Blood concentrations of total protein and albumin can be influenced by and indicative of protein nutrition and status. The lowest total protein concentration among periods for period 3 and a higher level of albumin in periods 2 and 3 vs. 1 suggest that levels of other proteins such as globulin varied as well.

In general, blood concentrations of NEFA, glucose, triglycerides, and cholesterol are related to energy metabolism, and a high level of NEFA can reflect tissue energy mobilization. That is, the concentration of NEFA increases when energy demands cannot be met from feed intake, with lipolysis occurring to supply precursor molecules for energy [59]. Factors responsible for the interaction between supplementation treatment and period in cholesterol and NEFA levels are not clearly understood. The NEFA concentration in period 1 did not differ between supplementation treatments, which might reflect the similar conditions before the experiment. Conversely, the greater level of NEFA in period 2 and 3 for LS than for HS could be due to greater lipolysis for LS in accordance with differences in nutrient and energy intake. This was also reflected in lower whole body fat mass for LS vs. HS. The greater blood cholesterol level for LS vs. HS in period 1 and 2 presumably is attributable to greater synthesis in the liver in response to relatively low energy status to support transport of triglycerides for an energy source [60]. It is unclear why the level of cortisol was greater for HS vs. LS, although the lower level in periods 2 and 3 than in period 1 may indicate adaptation to the nutritional planes and other experimental conditions.

### 4.5. BCS and BMI Relationships with Body Composition

Similar to body mass and composition data, all BMI, but not BCS except in wk 8, were affected by breed. This suggests that BMI can characterize genotype differences in whole body composition more accurately than BCS. In contrast to effects of supplementation treatment on whole body composition, BMI were not affected in wk 0, 4, or 8 but BCS was influenced in wk 8. However, there were tendencies for similar effects of supplementation treatment on two BMI at wk 8 as well. Differences in BCS in the weeks of this experiment between 2.9 and 3.6 appear to be reflected primarily in fat deposited subcutaneously [61,62]. In this range of BCS, fat appears mostly deposited subcutaneously [63], which could impact BCS relatively more than BMI.

Correlations between whole body fat concentration and BCS ranged from 0.53 to 0.63, which was relatively lower than in other studies with sheep such as Frutos et al. [61,64] and Sanson et al. [9]. This difference could be partially attributed to ones in nutritional planes and body composition perhaps smaller than in such other studies, with those in the current experiment again intended to reflect common and practical production settings encountered by commercial producers. Similar to findings of this experiment, Wang et al. [28] noted slightly stronger relationships between some BMI and performance measures compared with BCS. However, one difference is that Wang et al. [28] concluded that the four BMI determined in this experiment, based on wither height, heart girth, and length from the point of the shoulder to the hook and pin bones, were comparable. This is in contrast to a number of significant correlations between composition measures and BCS and BMI-GH that were of similar magnitude and lower than ones involving the other three BMI.

## 5. Conclusions

Supplementation treatment (fed at 0.16% of initial BW or a mixture of 25% soybean meal and 75% ground corn at 0.8% of initial BW) imposed before (71 to 85 days) and during breeding (17 days) did not influence reproductive performance of any of these breeds of hair sheep despite differences in BW, BCS, and body composition. This may be due to the moderate BCS (3 to 3.5) before breeding for both treatments, supplement treatments eliciting nutritional planes similar to ones commonly encountered by commercial producers, and increased straw intake with the LS supplementation treatment. Also, a BMI can be more highly related to and predictive of change in body composition of hair sheep resulting from different supplementation treatments compared with BCS.

## Figures and Tables

**Figure 1 animals-13-00735-f001:**
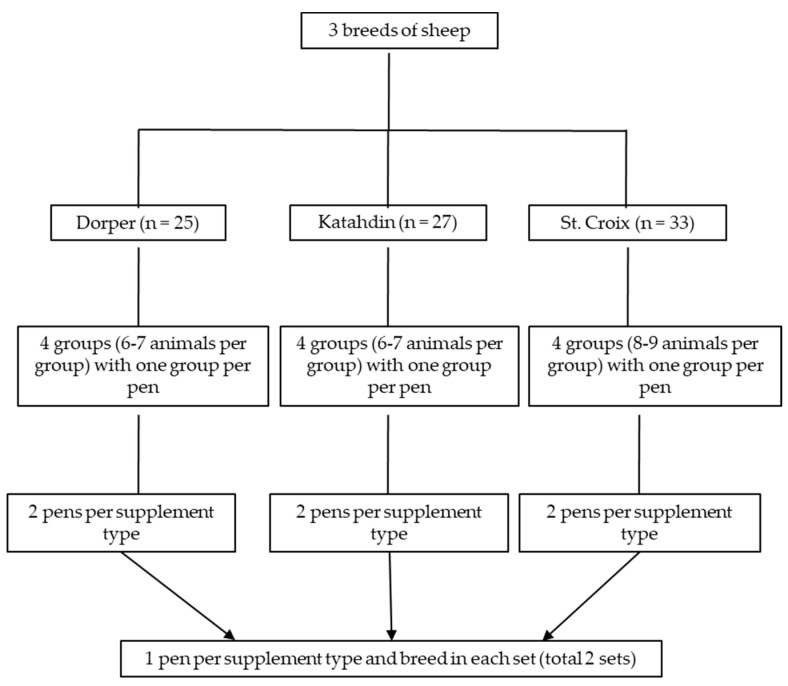
Schematic presentation of the experimental design.

**Figure 2 animals-13-00735-f002:**
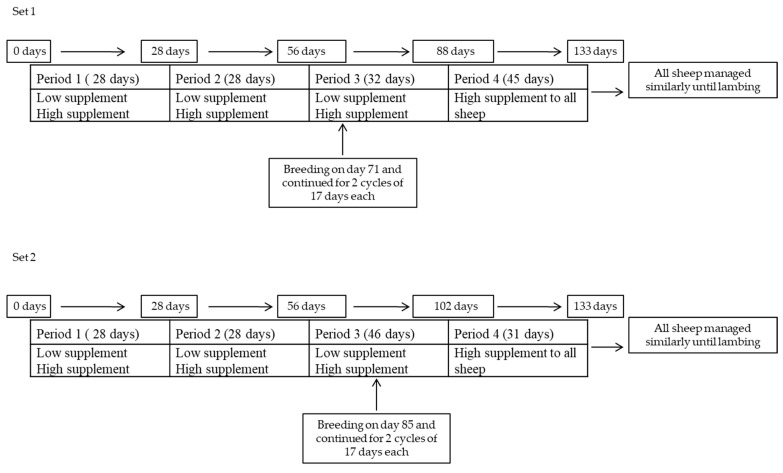
Two supplements fed to three breeds of sheep during different periods.

**Table 1 animals-13-00735-t001:** Composition of wheat straw and supplements (dry matter basis) ^1,2^.

	Wheat Straw		Low Supplement		High Supplement	
Item ^3^	Mean	SEM	Mean	SEM	Mean	SEM
Ash (%)	6.4	0.08	8.2	0.08	3.7	0.14
NDF (%)	81.9	0.59	11.0	0.19	13.2	0.35
ADF (%)	52.8	0.40	7.9	0.19	4.6	0.14
ADL (%)	9.9	0.16				
CP (%)	4.1	0.17	48.9	0.16	19.1	0.74

^1^ Low supplement = soybean meal; High supplement 25% soybean meal and 75% ground corn. ^2^ Mean of weekly samples. ^3^ NDF, neutral detergent fiber; ADF, acid detergent fiber; ADL, acid detergent lignin; CP, crude protein.

**Table 2 animals-13-00735-t002:** Effects of breed and supplementation treatment on dry matter intake by hair sheep ^1^.

	*p* Value ^2^			Dorper		Katahdin		St. Croix			
Item	Brd	Sup	Brd*Sup	Low	High	Low	High	Low	High	SEM	Brd ^3^
Period 1											
g/day											
Supplement	0.005	<0.001	0.065	109	544	116	542	88	460	11.3	S < D&K
Wheat straw	0.399	0.013	0.756	1005	783	1133	821	1107	920	83.6	
Total	0.580	0.044	0.832	1114	1326	1248	1363	1195	1380	82.2	
% body weight											
Supplement	0.873	<0.001	0.501	0.16	0.81	0.18	0.80	0.16	0.80	0.011	
Wheat straw	0.031	0.009	0.788	1.51	1.17	1.75	1.21	2.06	1.61	0.141	S > D&K
Total	0.031	0.139	0.731	1.67	1.99	1.92	2.01	2.23	2.41	0.139	S > D&K
Period 2											
g/day											
Supplement	0.018	<0.001	0.125	107	545	105	551	84	472	13.4	S < D&K
Wheat straw	0.749	0.022	0.318	1364	1013	1281	1130	1194	1096	79.6	
Total	0.442	0.018	0.433	1471	1558	1385	1681	1281	1568	85.0	
% body weight											
Supplement	0.203	<0.001	0.138	0.16	0.79	0.16	0.79	0.16	0.80	0.003	
Wheat straw	0.066	0.002	0.549	2.05	1.47	1.99	1.61	2.22	1.85	0.102	
Total	0.060	0.069	0.529	2.21	2.25	2.15	2.40	2.39	2.65	0.101	
Period 3											
g/day											
Supplement	0.005	<0.001	0.046	119 ^a^	558 ^c^	112 ^a^	565 ^c^	94 ^a^	475 ^b^	11.6	
Wheat straw	0.268	0.010	0.332	1515	1093	1391	1196	1246	1081	86.0	
Total	0.129	0.069	0.417	1634	15651	1503	1761	1340	1555	90.4	
% body weight											
Supplement	0.752	<0.001	0.626	0.18	0.80	0.18	0.80	0.18	0.80	0.005	
Wheat straw	0.217	<0.001	0.394	2.29	1.56	2.17	1.68	2.33	1.83	0.088	
Total	0.215	0.568	0.384	2.47	2.36	2.35	2.48	2.51	2.63	0.089	
Period 4 (g/day)											
Supplement	0.004	0.008	0.619	522	561	493	568	403	475	19.7	S < D&K
Wheat straw	0.358	0.020	0.746	1189	981	1222	1138	1055	994	100.2	
Total	0.175	0.580	0.712	1711	1542	1715	1706	1457	1468	115.9	

^1^ Supplementation treatments in periods 1, 2, and 3 were soybean meal fed at approximately 0.16% body weight and a mixture of 25% soybean meal and 75% ground corn given at approximately 0.8% body weight (dry matter basis; Low and High, respectively); periods 1 and 2 were each 4 wk in length; period 3 was 32 days for one pen of animals per treatment (set 1) and 46 days for the second pen (set 2); period 4 was 45 and 3 days for sets 1 and 2, respectively, during which time all animals were on the High supplement treatment; breeding began after 71 and 85 days on the different supplementation treatments for sets 1 and 2, respectively, with the different supplementation treatments imposed for an additional 17 days; the total length the different supplementation treatments were imposed was 88 and 102 days for sets 1 and 2, respectively. ^2^ Brd = breed; Sup = supplementation treatment.^3^ D = Dorper; K = Katahdin; S = St. Croix; main effect mean differences (*p* < 0.05) with nonsignificant interactions between breed and supplementation treatment (*p* > 0.05). ^a,b,c^ Means within a row without a common superscript letter differ (*p* < 0.05).

**Table 3 animals-13-00735-t003:** Effects of breed and supplementation treatment on body weight and body composition of hair sheep based on urea space measures ^1^.

		*p* Value ^2^			Dorper		Katahdin		St. Croix			
Item ^3^	n ^4^	Brd	Sup	Brd*Sup	Low	High	Low	High	Low	High	SEM	Brd ^5^
Initial												
BW, full (kg)	75	0.011	0.778	0.489	65.94	62.89	64.85	65.99	52.54	56.40	2.537	S < D&K
BW, shrunk (kg)	75	0.012	0.843	0.669	62.10	59.99	61.57	62.25	50.23	52.98	3.601	S < D&K
Water (kg)	75	0.005	0.885	0.615	29.13	28.53	28.94	28.83	24.74	25.73	0.801	S < D&K
Fat (kg)	75	0.026	0.814	0.686	22.59	21.34	22.32	23.14	17.02	18.34	1.241	S < D&K
Protein (kg)	75	0.005	0.885	0.615	7.91	7.75	7.86	7.83	6.72	6.99	0.218	S < D&K
Ash (kg)	75	0.005	0.885	0.615	1.85	1.81	1.84	1.83	1.57	1.64	0.051	S < D&K
Energy (MJ)	75	0.022	0.819	0.685	1071	1018	1060	1091	824.5	882.7	62.95	S < D&K
Water (%)	75	0.134	0.967	0.716	47.37	48.19	47.55	46.97	49.33	49.02	0.854	
Fat (%)	75	0.197	0.955	0.695	35.82	34.81	35.57	36.41	33.77	34.09	1.043	
Protein (%)	75	0.134	0.967	0.716	12.87	13.10	12.92	12.76	13.41	13.32	0.232	
Ash (%)	75	0.134	0.967	0.716	3.01	3.06	3.02	2.99	3.14	3.12	0.054	
Energy (MJ/kg)	75	0.208	0.954	0.691	17.06	16.72	16.98	17.27	16.38	16.49	0.357	
Week 4												
BW, full (kg)	79	0.012	0.187	0.973	67.14	70.62	64.97	68.10	54.44	58.88	3.017	S < D&K
BW, shrunk (kg)	79	0.015	0.109	0.980	61.87	66.22	59.09	64.26	50.43	54.50	2.939	S < D&K
Water (kg)	79	0.022	0.224	0.994	28.42	29.51	27.88	29.23	24.45	25.75	1.124	S < D&K
Fat (kg)	79	0.018	0.074	0.941	23.35	26.11	21.35	24.58	17.63	19.82	1.538	S < D&K
Protein (kg)	79	0.022	0.224	0.994	7.72	8.02	7.58	7.94	6.64	7.00	0.305	S < D&K
Ash (kg)	79	0.022	0.224	0.994	1.81	1.88	1.77	1.86	1.55	1.64	0.071	S < D&K
Energy (MJ)	79	0.016	0.077	0.950	1097	1212	1015	1150	846.8	941.1	65.87	S < D&K
Water (%)	79	0.138	0.179	0.911	46.54	45.26	47.70	46.08	48.59	47.80	0.989	
Fat (%)	79	0.209	0.204	0.895	37.04	38.57	35.53	37.57	34.82	35.69	1.270	
Protein (%)	79	0.138	0.179	0.911	12.65	12.30	12.96	12.52	13.20	12.99	0.269	
Ash (%)	79	0.138	0.179	0.911	2.96	2.88	3.03	2.93	3.09	3.04	0.063	
Energy (MJ/kg)	79	0.222	0.208	0.893	17.49	18.01	16.97	17.67	16.74	17.04	0.438	
Week 8												
BW, full (kg)	85	0.002	0.017	0.648	66.88	70.06	63.89	71.09	53.29	59.36	2.056	S < D&K
BW, shrunk (kg)	85	0.003	0.017	0.756	61.30	64.95	59.06	65.76	49.19	55.43	2.063	S < D&K
Water (kg)	85	0.003	0.033	0.871	28.66	29.95	28.19	30.27	24.39	26.37	0.790	S < D&K
Fat (kg)	85	0.003	0.010	0.632	22.45	24.25	20.90	24.60	16.49	19.86	0.977	S < D&K
Protein (kg)	85	0.003	0.033	0.871	7.79	8.14	7.66	8.23	6.63	7.17	0.215	S < D&K
Ash (kg)	85	0.003	0.033	0.871	1.82	1.90	1.79	1.92	1.55	1.68	0.050	S < D&K
Energy (MJ)	85	0.003	0.011	0.656	1063	1142	998.9	1157	801.5	946.5	42.99	S < D&K
Water (%)	85	0.015	0.014	0.724	47.24	46.30	48.16	46.43	49.70	48.01	0.520	S > D&K
Fat (%)	85	0.023	0.014	0.706	36.05	37.15	34.87	37.00	33.37	35.32	0.611	S < D
Protein (%)	85	0.015	0.014	0.724	12.84	12.58	13.09	12.62	13.50	13.05	0.141	S > D&K
Ash (%)	85	0.015	0.014	0.724	3.00	2.94	3.06	2.95	3.16	3.05	0.033	S > D&K
Energy (MJ/kg)	85	0.025	0.014	0.704	17.15	17.52	16.74	17.45	16.24	16.90	0.244	S < D
Change, wk 0 to 4												
BW, full (kg)	69	0.446	0.005	0.925	1.20	4.01	0.07	2.85	0.51	3.86	0.827	
BW, shrunk (kg)	69	0.439	0.002	0.601	−0.23	2.61	−2.39	2.48	−0.77	3.01	0.931	
Water (kg)	69	0.752	0.048	0.682	−0.71	−0.13	−1.07	0.54	−0.59	0.59	0.553	
Fat (kg)	69	0.453	0.032	0.956	0.77	2.76	−0.85	1.69	0.07	2.15	0.967	
Protein (kg)	69	0.752	0.048	0.682	−0.19	−0.04	−0.29	0.15	−0.16	0.16	0.150	
Ash (kg)	69	0.752	0.048	0.682	−0.04	−0.01	−0.07	0.03	−0.04	0.04	0.035	
Energy (MJ)	69	0.437	0.020	0.927	25.8	107.8	−40.2	69.8	−1.00	88.3	36.56	
Change, wk 4 to 8												
BW, full (kg)	79	0.421	0.023	0.266	−0.27	1.04	−1.35	1.93	−0.92	−0.10	0.724	
BW, shrunk (kg)	79	0.885	0.057	0.746	−0.58	0.24	−0.39	0.43	−1.00	0.52	0.547	
Water (kg)	79	0.699	0.202	0.892	0.24	1.09	0.26	0.64	0.00	0.49	0.488	
Fat (kg)	79	0.721	0.691	0.690	−0.91	−1.30	−0.75	−0.47	−0.99	−0.17	0.686	
Protein (kg)	79	0.699	0.202	0.892	−0.07	0.30	0.07	0.17	0.00	0.13	0.133	
Ash (kg)	79	0.699	0.202	0.892	0.02	0.07	0.02	0.04	0.00	0.03	0.031	
Energy (MJ)	79	0.735	0.554	0.674	−34.2	−44.3	−27.7	−14.6	−38.9	−3.5	24.99	
Change, wk 0 to 8												
BW, full (kg)	75	0.311	0.001	0.466	0.93	5.37	−1.17	4.94	−0.18	3.77	0.873	
BW, shrunk (kg)	75	0.700	<0.001	0.430	−0.80	2.90	−2.74	3.33	−2.02	3.36	0.842	
Water (kg)	75	0.954	0.001	0.387	−0.47	0.59	−0.89	1.13	−0.82	0.87	0.032	
Fat (kg)	75	0.660	0.008	0.850	−0.14	2.03	−1.45	1.70	−0.85	2.10	0.862	
Protein (kg)	75	0.954	0.001	0.387	−0.13	0.16	−0.24	0.31	−0.22	0.23	0.087	
Ash (kg)	75	0.954	0.001	0.387	−0.03	0.04	−0.06	0.07	−0.05	0.06	0.020	
Energy (MJ)	75	0.658	0.005	0.802	−8.4	83.7	−62.5	73.8	−38.4	88.1	33.30	

^1^ Based on measures at the beginning of the study and 4 and 8 wk later. Supplementation treatments were soybean meal fed at approximately 0.16% body weight and a mixture of 25% soybean meal and 75% ground corn given at approximately 0.8% body weight (dry matter basis; Low and High, respectively). ^2^ Brd = breed; Sup = supplementation treatment. ^3^ BW = body weight; shrunk BW was that after approximately 24 h without feed and water and was considered empty BW. ^4^ The number of individual animal observations, although animal group within breed and supplementation treatment was the experimental unit. ^5^ D = Dorper; K = Katahdin; S = St. Croix; main effect mean differences (*p* < 0.05) with nonsignificant interactions between breed and supplement treatment (*p* > 0.05).

**Table 4 animals-13-00735-t004:** Effects of breed and supplementation treatment on reproductive performance of hair sheep ^1^.

	*p* Value ^2^			Dorper		Katahdin		St. Croix			
Item ^3^	Brd	Sup	Brd*Sup	Low	High	Low	High	Low	High	SEM	Brd ^4^
Birth rate (%)	0.970	0.587	0.334	91.7	100.0	100.0	92.0	94.1	100.0	5.02	
Litter size	0.702	0.130	0.377	1.73	1.69	1.62	2.08	1.69	1.94	0.157	
Birth weight (kg)											
Individual lamb	0.048	0.118	0.804	4.35	4.15	4.66	4.13	3.87	3.53	0.236	S < K
Total litter	0.179	0.523	0.436	7.28	6.75	7.18	8.25	6.37	6.74	0.554	
Gestation length (days)	0.481	0.398	0.579	149.2	150.1	146.7	149.1	149.0	148.6	1.31	
Services (number) ^5^	0.100	0.691	0.069	1.18	1.54	1.08	1.08	1.31	1.06	0.104	

^1^ Supplementation treatments in periods 1, 2, and 3 were soybean meal fed at approximately 0.16% body weight and a mixture of 25% soybean meal and 75% ground corn given at approximately 0.8% body weight (dry matter basis; Low and High, respectively); periods 1 and 2 were each 4 wk in length; period 3 was 32 days for one pen of animals per treatment (set 1) and 46 days for the second pen (set 2); period 4 was 45 and 3 days for sets 1 and 2, respectively, during which time all animals were on the High supplementation treatment; breeding began after 71 and 85 days on the different supplementation treatments for sets 1 and 2, respectively, with the different supplementation treatments imposed for an additional 17 days; the total length the different supplementation treatments were imposed was 88 and 102 days for sets 1 and 2, respectively. ^2^ Brd = breed; Sup = supplement treatment. ^3^ Birth rate, litter size, and number of services were analyzed with the GLIMMIX procedure and individual lamb and total litter birth weights were analyzed with the MIXED procedure of SAS; gestation length was based on the last day of breeding observed and the date of birth. ^4^ K = Katahdin; S = St. Croix; main effect mean differences (*p* < 0.05) with nonsignificant interactions between breed and supplementation treatment (*p* > 0.05). ^5^ Number of services per conception.

**Table 5 animals-13-00735-t005:** *p* values for effects of breed, supplementation treatment, and period on heart rate and blood constituent concentrations in hair sheep ^1^.

	Source of Variation ^2^						
Item ^3^	Brd	Sup	Brd*Sup	Prd	Brd*Prd	Sup*Prd	Brd*Sup*Prd
Heart rate (beats/min)	0.855	0.076	0.506	0.208	0.646	0.897	0.066
Glucose (mg/dL)	0.877	0.434	0.933	0.955	0.684	0.843	0.980
Lactate (mg/dL)	0.058	0.151	0.647	0.002	0.118	0.418	0.747
Total protein (g/dL)	0.247	0.293	0.576	<0.001	0.607	0.428	0.926
Albumin (g/dL)	0.173	0.257	0.947	<0.001	0.684	0.321	0.952
Cholesterol (mg/dL)	0.098	0.134	0.757	<0.001	0.585	0.047	0.957
Triglycerides (mg/dL)	0.207	0.078	0.323	<0.001	0.394	0.051	0.049
NEFA (mEq/l)	0.263	0.001	0.974	<0.001	0.111	0.001	0.739
TAC (μM)	0.011	0.389	0.999	<0.001	0.006	0.010	0.002
Cortisol (ng/mL)	0.116	0.015	0.834	0.022	0.507	0.317	0.840

^1^ Periods 1 and 2 were 4 wk in length, with heart rate determined in the third week of each period (wk 3 and 7). Blood samples were collected at the end of periods 1, 2, and 3, with the latter being 32 and 46 days in length for sets 1 and 2, respectively. Supplementation treatments were soybean meal fed at approximately 0.16% body weight and a mixture of 25% soybean meal and 75% ground corn given at approximately 0.8% body weight (dry matter basis; Low and High, respectively). ^2^ Brd = breed; Sup = supplementation treatment; Prd = period. ^3^ NEFA = nonesterified fatty acids; TAC = Total antioxidant activity assessed by ferric reducing activity in plasma.

**Table 6 animals-13-00735-t006:** Effects of breed, supplementation treatment, and period on heart rate and blood constituent concentrations in hair sheep ^1^.

	Interaction ^2^		Breed ^3^				Sup			Period			
Item ^4^	Sup	Prd	DOR	KAT	STC	SEM	Low	High	SEM	1	2	3	SEM
HR (beats/min)			75.6	76.1	77.8	2.91	72.9	80.1	2.37	77.7	75.3		1.92
GLC (mg/dL)			58.6	58.6	59.8	1.81	58.1	59.9	1.48	59.0	58.9	59.1	1.18
LAC (mg/dL)			19.8	26.0	18.6	1.91	18.6	19.7	1.65	25.5 ^b^	20.9 ^a^	18.1 ^a^	1.63
TP (g/dL)			7.35	7.65	7.52	0.112	7.43	7.58	0.092	7.73 ^c^	7.53 ^b^	7.25 ^a^	0.082
ALB (g/dL)			2.65	2.55	2.64	0.035	2.59	2.64	0.028	2.53 ^a^	2.63 ^b^	2.68 ^b^	0.029
CHL (mg/dL)			82.1	77.7	74.5	2.05	80.2	76.1	1.67				
	Low									88.1 ^c^	80.3 ^b^	72.0 ^a^	2.32
	High									82.2 ^b^	73.4 ^a^	73.6 ^a^	
TG (mg/dL)			28.8	28.9	25.1	1.52	25.7	29.5	1.24	24.0 ^a^	30.9 ^c^	27.9 ^b^	1.11
	Low	1	22.8 ^ab^	25.0 ^abc^	22.2 ^a^	2.73							
	Low	2	31.5 ^cde^	25.8 ^abcd^	25.1 ^abc^								
	Low	3	26.2 ^abcd^	25.3 ^abc^	27.9 ^bcde^								
	High	1	24.4 ^abc^	26.7 ^abcd^	23.2 ^ab^								
	High	2	34.6 ^ef^	39.6 ^f^	29.1 ^bcde^								
	High	3	33.4 ^def^	31.1 ^cde^	23.2 ^ab^								
NEFA (mEq/L)			0.400	0.346	0.400								
	Low									0.546 ^c^	0.328 ^b^	0.503 ^c^	0.0347
	High									0.513 ^c^	0.198 ^a^	0.203 ^a^	
TAC (μM)			200 ^a^	221 ^b^	219 ^b^	3.6	211	215	2.9	199 ^a^	215 ^b^	225 ^c^	3.1
	Low	1	183 ^a^	209 ^bc^	191 ^a^	7.5							
	Low	2	201 ^ab^	217 ^bcd^	209 ^bc^								
	Low	3	210 ^bcd^	230 ^de^	250 ^e^								
	High	1	186 ^a^	204 ^ab^	219 ^bcd^								
	High	2	201 ^ab^	248 ^e^	216 ^bcd^								
	High	3	217 ^bcd^	216 ^bcd^	227 ^cd^								
COR (ng/mL)			19.9	28.6	25.5	2.42	20.0 ^a^	29.4 ^b^	1.97	28.8 ^b^	22.6 ^a^	22.7 ^a^	2.02

^1^ Periods 1 and 2 were 4 wk in length, with heart rate in the third week of each period (wk 3 and 7). Blood samples were collected at the end of periods 1, 2, and 3, with the latter being 32 and 46 days in length for sets 1 and 2, respectively. Supplementation treatments were soybean meal fed at approximately 0.16% body weight and a mixture of 25% soybean meal and 75% ground corn given at approximately 0.8% body weight (dry matter basis; Low and High, respectively). ^2^ Sup = supplementation treatment; Prd = period. ^3^ DOR = Dorper; KAT = Katahdin; STC = St. Croix. ^4^ HR = heart rate; GLC = glucose; LAC = lactate; TP = total protein; ALB = albumin; CHL = cholesterol; TG = triglycerides; TAC = Total antioxidant activity assessed by ferric reducing activity in plasma; COR = cortisol. ^a,b,c,d,e,f^ Means within grouping without a common superscript letter differ (*p* < 0.05).

**Table 7 animals-13-00735-t007:** Effects of breed and supplementation treatment on body condition score and body mass indexes of hair sheep ^1^.

		*p* Value ^2^			Dorper		Katahdin		St. Croix		
Item ^3^	n ^4^	Brd	Sup	Brd*Sup	Low	High	Low	High	Low	High	SEM
Initial											
BCS	85	0.211	0.397	0.772	3.29	3.41	3.29	3.29	2.99	3.18	0.139
BMI-WH	85	0.005	0.579	0.735	16.13	16.96	14.88	14.96	13.44	13.38	0.586
BMI-WP	85	0.005	0.623	0.847	13.72	14.22	12.69	12.81	11.53	11.50	0.463
BMI-GH	85	0.021	0.840	0.693	10.90	11.31	10.43	10.35	9.84	9.69	0.339
BMI-GP	85	0.020	0.914	0.800	9.27	9.49	8.89	8.86	8.44	8.33	0.258
Week 4											
BCS	85	0.499	0.108	0.634	3.27	3.36	3.29	3.46	3.06	3.39	0.129
BMI-WH	85	0.049	0.127	0.801	16.15	17.38	15.25	15.80	13.53	15.09	0.770
BMI-WP	85	0.030	0.112	0.762	13.84	14.74	13.03	13.43	11.70	12.90	0.548
BMI-GH	85	0.093	0.347	0.745	11.26	11.70	10.95	10.94	9.99	10.65	0.433
BMI-GP	85	0.052	0.368	0.661	9.65	9.93	9.35	0.30	8.64	9.11	0.282
Week 8											
BCS	85	0.042	0.009	0.647	3.27	3.61	3.33	3.57	2.91	3.36	0.111
BMI-WH	85	0.010	0.092	0.943	15.65	16.91	14.53	15.38	12.48	13.76	0.690
BMI-WP	85	0.015	0.083	0.955	13.46	14.52	12.50	13.30	10.96	12.09	0.586
BMI-GH	85	0.019	0.288	0.873	10.86	11.05	10.23	10.57	9.12	9.71	0.391
BMI-GP	85	0.032	0.255	0.853	9.33	9.49	8.80	9.13	8.01	8.53	0.326
Change, wk 0 to 4											
BCS	85	0.124	0.147	0.380	−0.02	−0.05	0.00	0.18	0.06	0.20	0.071
BMI-WH	85	0.272	0.035	0.246	0.025	0.426	0.369	0.846	0.089	1.710	0.375
BMI-WP	85	0.480	0.083	0.412	0.118	0.517	0.338	0.623	0.167	1.401	0.375
BMI-GH	85	0.822	0.278	0.409	0.367	0.395	0.522	0.593	0.147	0.959	0.312
BMI-GP	85	0.944	0.300	0.371	0.378	0.447	0.462	0.433	0.194	0.782	0.225
Change, wk 4 to 8											
BCS	85	0.025	0.026	0.387	0.00	0.25	0.04	0.11	−0.15	−0.03	0.060
BMI-WH	85	0.417	0.972	0.865	−0.504	−0.477	−0.726	−0.412	−1.049	−1.329	0.554
BMI-WP	85	0.527	0.688	0.876	−0.383	−0.219	−0.524	−0.132	−0.739	−0.814	0.463
BMI-GH	85	0.566	0.981	0.764	−0.396	−0.649	−0.717	−0.375	−0.870	−0.935	0.394
BMI-GP	85	0.673	0.690	0.712	−0.315	−0.446	−0.555	−0.163	−0.627	−0.575	0.303
Change, wk 0 to 8											
BCS	85	0.288	0.006	0.963	−0.02	0.20	0.04	0.29	−0.08	0.17	0.070
BMI-WH	85	0.791	0.090	0.677	−0.478	−0.051	−0.357	0.424	−0.960	0.381	0.514
BMI-WP	85	0.947	0.093	0.806	−0.265	0.297	−0.185	0.491	−0.571	0.587	0.488
BMI-GH	85	0.396	0.181	0.239	−0.038	−0.254	−0.194	0.219	−0.723	0.025	0.254
BMI-GP	85	0.642	0.135	0.346	0.062	0.000	−0.092	0.270	−0.433	0.206	0.221

^1^ Based on measures at the beginning of the study and 4 and 8 wk later; supplementation treatments were soybean meal fed at approximately 0.16% body weight and a mixture of 25% soybean meal and 75% ground corn given at approximately 0.8% body weight (dry matter basis; Low and High, respectively). ^2^ Brd = breed; Sup = supplementation treatment. ^3^ BCS = body condition score in a 5-point scale; BMI = body mass index; Wither = height at withers; Hook = length from the point of the shoulder to hook bone; Pin = length from the point of the shoulder to pin bone; Heart = circumference of heart girth; BMI-WH = BW/(Wither × Hook) [g/cm^2^]; BMI-WP = BW/(Wither × Pin) [g/cm^2^]; BMI-GH = BW/(Heart × Hook) [g/cm^2^]; BMI-GP = BW/(Heart × Pin) [g/cm^2^]. ^4^ The number of individual animal observations, although animal group within breed and supplement treatment was the experimental unit.

**Table 8 animals-13-00735-t008:** Pearson correlation coefficient (r) between body condition score, body mass indexes, body weight, and body composition of hair sheep based on urea space measures at the beginning of the experiment and after 4 and 8 wk ^1^.

				Variable ^1^						
Week	n	Variable	Parameter	BCS	BW, Full (kg)	BW, Shrunk (kg)	Water (%)	Fat (%)	Protein (%)	Energy (MJ/kg)
0	75	BCS	r		0.70	0.69	−0.65	0.63	−0.65	0.63
			P		<0.001	<0.001	<0.001	<0.001	<0.001	<0.001
	75	BMI-WH	r	0.69	0.82	0.82	−0.72	0.70	−0.72	0.69
			P	<0.001	<0.001	<0.001	<0.001	<0.001	<0.001	<0.001
	75	BMI-WP	r	0.71	0.86	0.85	−0.77	0.74	−0.77	0.73
			P	<0.001	<0.001	<0.001	<0.001	<0.001	<0.001	<0.001
	75	BMI-GH	r	0.61	0.76	0.75	−0.64	0.61	−0.64	0.61
			P	<0.001	<0.001	<0.001	<0.001	<0.001	<0.001	<0.001
	75	BMI-GP	r	0.64	0.82	0.81	−0.70	0.67	−0.70	0.67
			P	<0.001	<0.001	<0.001	<0.001	<0.001	<0.001	<0.001
4	79	BCS	r		0.69	0.71	−0.57	0.53	−0.57	0.53
			P		<0.001	<0.001	<0.001	<0.001	<0.001	<0.001
	79	BMI-WH	r	0.73	0.89	0.89	−0.64	0.58	−0.64	0.57
			P	<0.001	<0.001	<0.001	<0.001	<0.001	<0.001	<0.001
	79	BMI-WP	r	0.72	0.90	0.90	−0.69	0.63	−0.69	0.63
			P	<0.001	<0.001	<0.001	<0.001	<0.001	<0.001	<0.001
	79	BMI-GH	r	0.64	0.85	0.84	−0.57	0.51	−0.57	0.50
			P	<0.001	<0.001	<0.001	<0.001	<0.001	<0.001	<0.001
	79	BMI-GP	r	0.62	0.86	0.85	−0.63	0.58	−0.63	0.57
			P	<0.001	<0.001	<0.001	<0.001	<0.001	<0.001	<0.001
8	85	BCS	r		0.73	0.75	−0.64	0.60	−0.64	0.59
			P		<0.001	<0.001	<0.001	<0.001	<0.001	<0.001
	85	BMI-WH	r	0.71	0.81	0.80	−0.68	0.63	−0.68	0.63
			P	<0.001	<0.001	<0.001	<0.001	<0.001	<0.001	<0.001
	85	BMI-WP	r	0.73	0.84	0.83	−0.70	0.66	−0.65	0.59
			P	<0.001	<0.001	<0.001	<0.001	<0.001	<0.001	<0.001
	85	BMI-GH	r	0.60	0.76	0.76	−0.65	0.60	−0.65	0.59
			P	<0.001	<0.001	<0.001	<0.001	<0.001	<0.001	<0.001
	85	BMI-GP	r	0.65	0.82	0.82	−0.70	0.65	−0.70	0.64
			P	<0.001	<0.001	<0.001	<0.001	<0.001	<0.001	<0.001

^1^ BCS = body condition score in a 5-point scale; BW = body weight; BMI = body mass index; Wither = height at withers; Hook = length from the point of the shoulder to hook bone; Pin = length from the point of the shoulder to pin bone; Heart = circumference of heart girth; BMI-WH = BW/(Wither × Hook) [g/cm^2^]; BMI-WP = BW/(Wither × Pin) [g/cm^2^]; BMI-GH = BW/(Heart × Hook) [g/cm^2^]; BMI-GP = BW/(Heart × Pin) [g/cm^2^].

**Table 9 animals-13-00735-t009:** Pearson correlation coefficient (r) between changes during 4 or 8 wk in body condition score, body mass indexes, body weight, and body composition of hair sheep based on urea space.

				Variable ^1^						
Week	n	Variable	Parameter	BCS	BW, Full (kg)	BW, Shrunk (kg)	Water (kg)	Fat (kg)	Protein (kg)	Energy (MJ)
0–4	69	BCS	r		0.38	0.34	0.20	0.17	0.20	0.20
			P		0.001	0.005	0.096	0.172	0.096	0.106
	69	BMI-WH	r	0.39	0.53	0.41	0.31	0.15	0.31	0.19
			P	0.001	<0.001	<0.001	0.010	0.234	0.010	0.127
	69	BMI-WP	r	0.44	0.61	0.51	0.18	0.37	0.18	0.41
			P	<0.001	<0.001	<0.001	0.132	0.002	0.132	0.001
	69	BMI-GH	r	0.22	0.33	0.18	0.35	−0.13	0.35	−0.11
			P	0.067	0.006	0.128	0.004	0.271	0.004	0.410
	69	BMI-GP	r	0.28	0.42	0.29	0.23	0.08	0.23	0.11
			P	0.022	<0.001	0.017	0.055	0.502	0.055	0.361
4–8	79	BCS	r		0.30	0.25	0.08	0.10	0.08	0.12
			P		0.008	0.025	0.494	0.369	0.494	0.300
	79	BMI-WH	r	0.22	0.39	0.17	−0.09	0.19	−0.09	0.19
			P	0.051	0.001	0.142	0.428	0.100	0.428	0.092
	79	BMI-WP	r	0.33	0.44	0.10	−0.12	0.17	−0.12	0.17
			P	0.003	<0.001	0.365	0.282	0.132	0.282	0.131
	79	BMI-GH	r	0.05	0.32	0.16	−0.15	0.23	−0.15	0.23
			P	0.662	0.004	0.162	0.202	0.044	0.202	0.042
	79	BMI-GP	r	0.14	0.36	0.09	−0.21	0.23	−0.21	0.23
			P	0.233	0.001	0.456	0.066	0.041	0.066	0.044
0–8	75	BCS	r		0.51	0.45	0.44	0.35	0.44	0.35
			P		<0.001	0.001	0.001	0.004	<0.001	0.002
	75	BMI-WH	r	0.29	0.57	0.56	0.39	0.52	0.39	0.54
			P	0.010	<0.001	<0.001	<0.001	<0.001	<0.001	<0.001
	75	BMI-WP	r	0.27	0.64	0.59	0.42	0.54	0.42	0.56
			P	0.020	<0.001	<0.001	<0.001	<0.001	<0.001	<0.001
	75	BMI-GH	r	0.24	0.39	0.37	0.32	0.31	0.32	0.32
			P	0.041	0.001	0.001	0.006	0.007	0.006	0.005
	75	BMI-GP	r	0.22	0.48	0.42	0.37	0.34	0.37	0.36
			P	0.060	<0.001	<0.001	0.001	0.003	0.001	0.002

^1^ BCS = body condition score in a 5-point scale; BW = body weight; BMI = body mass index; Wither = height at withers; Hook = length from the point of the shoulder to hook bone; Pin = length from the point of the shoulder to pin bone; Heart = circumference of heart girth; BMI-WH = BW/(Wither × Hook) [g/cm^2^]; BMI-WP = BW/(Wither × Pin) [g/cm^2^]; BMI-GH = BW/(Heart × Hook) [g/cm^2^]; BMI-GP = BW/(Heart × Pin) [g/cm^2^].

## Data Availability

Data are presented in the tables.

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
