# Peer review of "Effects of Nutritional Plane at Breeding on Feed Intake, Body Weight, Condition Score, Mass Indexes, and Chemical Composition, and Reproductive Performance of Hair Sheep"

_animals, 2023, doi:10.3390/ani13040735_

Round 1
Reviewer 1 Report
The experiment addresses the impact of supplementation strategies around the time of breeding on reproductive outcomes, body weight and conditions changes, blood parameters. It is not clear why the project was broken down into different phases, especially as these appear arbitrary and did not necessarily related to physiological stages (pre- vs. post breeding). It would be helpful to expand on why these experimental timelines were selected.
The following points may want to be addressed by the authors (as there was no line numbering for this manuscript it was not easy to identify the locations related to the comment):
Materials and Methods
‘Most of the sheep were ewes, but some had not previously given birth’. Sentence needs to be reworded, indicating the actual age structure of the flock and the parity status. The age structure is discussed later, so maybe the sentences should be deleted. Also, the age structure cited later suggests that there were ewes >2 years of age that were nulliparous? The wide range in ages (3 to 11 years may have influenced the absence of differences in reproductive performance.
The previous history of experimental use of the flock prior to the study is not relevant here and should be deleted, as it also creates an issue with self-citing.
Units (meters?) are needed for description of pen size.
The estrus synchronization protocol should be described in relation to the overall experimental time line; day 0, 9, and 10 should be converted to the appropriate day of the entire study.
Were separate colors used in marking harnesses to identify individual rams for the contribution to the breeding, to ensure that multiple rams are represented for each breed in the breeding outcomes?
Restrict references for body composition analysis to sheep references, unless a goat reference provides a unique aspect to the technique. This again causes issues with self-citing.
As heat energy could not be determined, descriptions of this effort should be deleted, as should be the references describing this technique for goats.
For the statistical analysis it should be stated that ewe, rather than animal group, was the experimental unit (one would assume).
Results
Orientation of table 2 to landscape format will greatly improve visualizing the data.
Table 2: if supplement level (intake) is set as part of the experimental design and no refusal was noted, why is there a need to include it as variable being analyzed, especially as part of % body weight?
Table 4 includes the variable ‘number of services’; how was this measured and how was it defined?
Discussion
In regard to reproductive performance it would be useful to have the discussion focus on characteristics likely impacted by the timing of this study (pre- and post-breeding), and why there was no effect on conception rate and litter size.
Author Response
The experiment addresses the impact of supplementation strategies around the time of breeding on reproductive outcomes, body weight and conditions changes, blood parameters. It is not clear why the project was broken down into different phases, especially as these appear arbitrary and did not necessarily related to physiological stages (pre- vs. post breeding). It would be helpful to expand on why these experimental timelines were selected.
Response: We have presented feed intake and blood variables in different periods after the nutritional planes were imposed. The nutritional effects on these variables important to understand the changes affected by treatment during the different periods rather than overall mean effect.
The following points may want to be addressed by the authors (as there was no line numbering for this manuscript it was not easy to identify the locations related to the comment):
Materials and Methods
‘Most of the sheep were ewes, but some had not previously given birth’. Sentence needs to be reworded, indicating the actual age structure of the flock and the parity status. The age structure is discussed later, so maybe the sentences should be deleted. Also, the age structure cited later suggests that there were ewes >2 years of age that were nulliparous? The wide range in ages (3 to 11 years may have influenced the absence of differences in reproductive performance.
Response: We have deleted the sentence ‘Most of the sheep were ewes, but some had not previously given birth’. There were only 4 primiparous - 1 was STC, 2 were KAT, and 1 was DOR. The animals were allocated based on initial age in two treatments.
The previous history of experimental use of the flock prior to the study is not relevant here and should be deleted, as it also creates an issue with self-citing.
Response: We have deleted the previous experiment information.
Units (meters?) are needed for description of pen size.
Response: Now the unit is mentioned.
The estrus synchronization protocol should be described in relation to the overall experimental time line; day 0, 9, and 10 should be converted to the appropriate day of the entire study.
Response: We have now also mentioned with respect to experimental time.
Were separate colors used in marking harnesses to identify individual rams for the contribution to the breeding, to ensure that multiple rams are represented for each breed in the breeding outcomes?
Response: Yes, we mentioned “Four rams of each breed divided into two sets were used, which were previously subjected to and passed a breeding soundness examination” and “time two rams of each breed fitted with marking harnesses were introduced into one of the two pens per breed-supplement treatment.”
Restrict references for body composition analysis to sheep references, unless a goat reference provides a unique aspect to the technique. This again causes issues with self-citing.
Response: We have deleted goat references and restricted sheep references.
As heat energy could not be determined, descriptions of this effort should be deleted, as should be the references describing this technique for goats.
Response: We have removed the description of heat energy from method section, but we have introduced in the discussion section as heart rate had a tendency to increase due to high supplementation and it is likely due to increased total feed intake.
For the statistical analysis it should be stated that ewe, rather than animal group, was the experimental unit (one would assume).
Response: The experimental unit was pen or animal group within breed x supplement treatment as is stated.
Results
Orientation of table 2 to landscape format will greatly improve visualizing the data.
Response: The formatting has been altered during formatting as per the styles of the journal by the editorial office. We have formatted the tables properly and finally we will check in the proof.
Table 2: if supplement level (intake) is set as part of the experimental design and no refusal was noted, why is there a need to include it as variable being analyzed, especially as part of % body weight?
Response: We agree with the reviewers that as there was no supplement refusal, supplement intake as %BW is not highly important. For this reason, we did not describe supplement intake in the result section. However, we thought that readers could easily understand the intake information from the table if the information is available together.
Table 4 includes the variable ‘number of services’; how was this measured and how was it defined?
Response: it has now been defined as a footnote to the table and in the texts.
Discussion
In regard to reproductive performance it would be useful to have the discussion focus on characteristics likely impacted by the timing of this study (pre- and post-breeding), and why there was no effect on conception rate and litter size.
Response: It is good point. We have now discussed based on the suggestion in the discussion section.

Reviewer 2 Report
In general, it is well written, despite possibly not obtaining the expected results, it is still precious results for breeders of these breeds of sheep.
The article is a little long and it is possible to improve some aspects, which I refer to in the attached file.

Author Response
General comments
In general, it is well written, despite possibly not obtaining the expected results, it is still precious results for breeders of these breeds of sheep. The article is a little long and it is possible to improve some aspects, which I refer to in the following points.
Response: thank you for your suggestions.
Abstract
The abstract is too long, according to the instructions, it should not exceed 200 words. From what was possible to ascertain, the present abstract exceeds 500 words.
Response: We agree the abstract is descriptive, but we believe that readers can get better idea about the findings here what are presented in the manuscript. We have noticed that Animals journal do not highly restrict the abstract word limit.
keywords
In my opinion, some keywords are not the most suitable. I suggest you review the keywords since they must be different from the title.
Response: All the keywords are relevant in this study. This journal does not state that keywords should be different what is presented in abstract and title.
Introduction
The objectives of the abstract are more explicit than in the introduction. Rewrite the objectives of the work and define the general and specific objectives.
Response: The objectives stated in abstract and introduction section are usually similar. Nonetheless, we have deleted BW and used BCS here as BCS is more important for the reproductive performances.
Material and methods
In point 2.2 the materials and methods must be exact in the protocol so that the test can be replicated, avoiding expressions such as approximately. for example in the sentence "Approximately 1 month before lambing, we were vaccinated again against clostridial organisms (...) , must indicate the exact days, with the possibility of placing a deviation in front. You must also avoid abbreviations, such as "wk". These inaccuracies must be corrected.
Response: Stating approximately 1 month seems adequate since the exact lambing date was not known, and effects of vaccination would not be expected to be markedly affected by a few days of variation.
Point 2.6 The statistical analysis should be more detailed, indicating which tests were used, whether or not the data are parametric, whether they have a normal distribution...
Response: A couple of sentences have been added, one to address the normality of data distribution and another covariance structure. Pages 71 and 72 in the smallholder handbook have a couple of quotes that could be considered for inclusion, but probably not needed.
Results The data in table 2 are confusing, the table should be restructured in order to make its interpretation clearer. The same applies to table 3, they should be more careful in presenting the data. Highlighting the weeks in bold, placing all the units in front of each parameter, in the title of the table the number 1 must be higher than the line, among other details make the difference to improve the reading of the article.
Response: We agree with the reviewer that formatting of the table was an issue which happened due to preparation of this manuscript as per the template of Animals by the editorial office. We have tried to reformat it and we will check further in the proof.
Part of the tables are statistical analysis, they should only be a supplement to the article.
Response: It is quite common to use main effect and interaction p-values in manuscript and they are a part of the tables. In Table 5, p-values are presently only as we could not accommodate it in Table 6. We believe the statistical information used side by side will be useful to understand the presentation of data in Table 6.
The results should be presented in a more summarized form.
Response: More specific suggestion of the reviewer is appreciated in this matter.
The discussion and the conclusion are fine, in the conclusion, attention should be paid to responding clearly to the objectives of the work, clearly indicating that the meal plans had no influence on the evaluated parameters.
Response: We have revised the conclusion section.

Round 2
Reviewer 2 Report
The power of synthesis is very important in the scientific world, so I insist that the abstract should be shorter than it actually is.
In the instructions for authors at: https://www.mdpi.com/journal/animals/instructions
in the abstract point, the first sentence says the following "Abstract: The abstract should be a total of about 200 words maximum. " If it was up to 250 words, sometimes it's possible, but in this case, it's more than 500 words. Review the abstract.
Some of the numbers in the table captions are wrong. the number above the line should be above the abbreviations, not in general. For example, in table 7, the number 2 is in the p-value, when we go to the legend the number 2 indicates Brd = breed; Sup = supplementation treatment.
Author Response
The changes in the texts are highlighted in yellow.
The power of synthesis is very important in the scientific world, so I insist that the abstract should be shorter than it actually is. In the instructions for authors at: https://www.mdpi.com/journal/animals/instructions
in the abstract point, the first sentence says the following "Abstract: The abstract should be a total of about 200 words maximum. " If it was up to 250 words, sometimes it's possible, but in this case, it's more than 500 words. Review the abstract.
Response: Thanks for your suggestion. We have reduced the word number in the abstract from 534 to 356. I hope this is acceptable now, but if the number needs to be reduced further we can do it.
Some of the numbers in the table captions are wrong. the number above the line should be above the abbreviations, not in general. For example, in table 7, the number 2 is in the p-value, when we go to the legend the number 2 indicates Brd = breed; Sup = supplementation treatment.
Response: We checked this issue, and it is alright to our opinion. For example, in Table 7, abbreviations used for the P value columns are Brd = breed; Sup = supplementation treatment. Otherwise, if we denote superscript number for each abbreviation, the number list will be lengthy. Nonetheless, if it is needed, the Copy Editor can perform it as per the need of this journal.
